# The Effect of the Immunization Schedule and Antibody Levels (Anti-S) on the Risk of SARS-CoV-2 Infection in a Large Cohort of Healthcare Workers in Northern Italy

**DOI:** 10.3390/vaccines11040746

**Published:** 2023-03-28

**Authors:** Emanuele Sansone, Giulia Collatuzzo, Stefano Renzetti, Giorgia Ditano, Carlo Bonfanti, Emma Sala, Luigina Terlenghi, Alberto Matteelli, Mahsa Abedini, Shuffield Seyram Asafo, Paolo Boffetta, Giuseppe De Palma

**Affiliations:** 1Unit of Occupational Health and Industrial Hygiene, Department of Medical and Surgical Specialties, Radiological Sciences and Public Health, University of Brescia, 25123 Brescia, Italy; 2Department of Medical and Surgical Sciences, University of Bologna, 40138 Bologna, Italy; 3Institute of Microbiology, Department of Molecular and Translational Medicine, University of Brescia and ASST Spedali Civili, 25123 Brescia, Italy; 4Unit of Occupational Health, Hygiene, Toxicology and Prevention, University Hospital ASST Spedali Civili, 25123 Brescia, Italy; 5Department of Infectious and Tropical Diseases, University of Brescia and ASST Spedali Civili, 25123 Brescia, Italy; 6Stony Brook Cancer Center, Stony Brook University, Stony Brook, NY 11794, USA

**Keywords:** SARS-CoV-2, HCWs, COVID-19 vaccines, immunization profiles, hybrid immunization, serological response

## Abstract

Given their occupational risk profile, HCWs were the first to receive anti-SARS-CoV-2 vaccination. However, breakthrough infections remained common, mainly sustained by new SARS-CoV-2 variants of concern (VOCs) that rapidly spread one after another in Italy. Evidence suggests that the measured level of anti-SARS-CoV-2 antibodies does not clearly predict the level of protection conferred by either natural infection or vaccine-induced immunization, highlighting the need for further study on the diversity in susceptibility to SARS-CoV-2 infection. The present study aimed to characterize different risk profiles for SARS-CoV-2 infection in HCWs who had recently received the booster dose, and who were classified according to their immunization profile. The very small number of workers infected during the 8 months following the primary-cycle administration represents proof of the vaccine’s effectiveness against non-omicron strains. The comparison among different immunization profiles showed that hybrid immunization (vaccine plus natural infection) elicits higher antibody levels. However, hybrid immunization does not always provide better protection against reinfection, thus suggesting that the immunization profile plays a major role as a virus–host interaction modifier. Despite the high resistance to the reinfection, the peri-booster infection had a not-neglectable infection rate (5.6%), this further reinforcing the importance of preventive measures.

## 1. Introduction

The COVID-19 pandemic represents one of the major events of the modern era, rapidly spreading from China to Europe and soon reaching a global dimension. Northern Italy, especially the Lombardy region, was among the most burdened areas of Europe in the early phase of the pandemic and had to face the spread of COVID-19 cases before other areas [1]. The Italian government declared the quarantine of 11 municipalities in Northern Italy on the 21st February, then this extended to the entire Lombardy region on the 8th March and finally to the whole country the next day [1].

In April 2020, more than 4000 cases per day were reported in Italy. In Lombardy, more than 560,000 deaths had occurred by 15 May 2020, with 25,782 excess deaths, the highest rate registered in the country [1,2]. Healthcare workers (HCWs) were consequently subjected to the challenge of a new virus, an overload of patients to manage, and a higher risk of infection [3]. Based on a meta-analysis of 97 publications from 2020, the overall prevalence of SARS-CoV-2 infection in HCWs was 11% according to PCR test results and 7% according to serological test results [3]. However, in Northern Italy, higher infection rates have been reported [4,5].

The introduction of newly developed anti-SARS-CoV-2 vaccines led to large changes in COVID-19 epidemiology, including a dramatic reduction in hospitalized and life-threatening cases, as well as deaths [6,7,8]. Given their occupational risk profile, HCWs were the first to receive anti-SARS-CoV-2 vaccination starting on 27 December 2020. In May 2021, when vaccination against SARS-CoV-2 became mandatory for Italian HCWs, the vast majority of them were vaccinated [5]. However, breakthrough infections remained common [9], mainly sustained by the spread of new SARS-CoV-2 variants of concern (VOCs) that rapidly spread one after another in our country: alpha (B.1.1.7 and B.1.1.7+E484K), gamma (P.1), delta (B.1.617.2), and omicrons (B.1.1.529) [10]. Compared to the very first COVID-19 outbreak, the rhythm of infection-spread appeared to increase logarithmically, with the Data Repository for COVID-19 Infection of the Center for Systems Science and Engineering (CSSE) reporting more than 200,000 new cases per day in January 2022, when the peak incidence was registered for 2022 [2]. This can be explained by the interplay of different factors, including the spread of omicron variants of SARS-CoV-2 and the increasing number of diagnostic tests performed once they became easily accessible [11,12,13].

The literature reports solid evidence of the decline in the severity of COVID-19 symptoms in vaccinated persons who were infected [9,11,14]. Comparison of serological test results of vaccinated individuals demonstrated higher antibody levels in cases of SARS-CoV-2 infection. In fact, natural infection acts as a booster for antibody development, leading to an enhanced serological response, which may confer higher protection against further infection. Thus, in the case of recent infection, many public health institutions recommended deferring booster doses [15].

The measured level of anti-SARS-CoV-2 antibodies does not clearly predict the level of protection conferred by either natural infection or vaccine-induced immunization [16]. Indeed, the diversity in susceptibility to SARS-CoV-2 infection in both vaccinated and unvaccinated individuals deserves further study [17,18].

One of the first studies reporting breakthrough infections in HCWs showed that cases had a lower level of peri-infection antibodies than matched uninfected controls [9]. An analysis within the large ORCHESTRA multicentric cohort study, to which the Brescia cohort belongs, reported an inverse relationship between serology level and risk of breakthrough infection [19].

In the present study, we aimed to characterize different risk profiles for SARS-CoV-2 infection assessed by rhino-pharyngeal swabs (RPS) in HCWs from Brescia using data gathered at the Occupational Medicine Unit during the epidemiological surveillance from March 2020 to October 2022. Our main objective was to provide new information on the determinants of the risk of SARS-CoV-2 19 infection in HCWs, who were classified according to serological test results into those with no evidence of infection and those who seroconverted and accounting for the timing of seroconversion in relation to three main periods: (i) before vaccination; (ii) between the first vaccine dose and approximately 8 months after the first dose; and (iii) more than 8 months after the first dose.

## 2. Materials and Methods

### 2.1. Study Design

We previously reported the main results obtained in the longitudinal follow-up of anti-S antibody levels observed among 7411 HCWs at 2, 4, and 8 months after the first dose of vaccine administration [5]. A further serological-screening campaign (here indicated as T_4_) was later performed in our hospital at 13–15 months after the initial dose of vaccine in January 2021 (see graphical abstract). Overall, our sero-surveillance programme lasted for almost 2 years, from April 2020 to March 2022; the results collected through seven serological-screening campaigns allowed us to monitor the immunological response to the primary cycle and to the booster dose and to define the immunization profile of each worker with optimal reliability. Blood samples collected during the post-vaccine serological screenings (from T_1_ up to T_4_) were tested for both quantitative anti-N and anti-S antibody levels.

We hypothesized that individual immunization profiles derived from SARS-CoV-2 infection and/or vaccine administration, which impact the observed anti-S antibody levels, can modify the risk of infection after booster-dose administration. The observational period started from the T_4_ sample collection date and ended on 31 October 2022. Only fully vaccinated workers (three doses) who had a baseline test (T_0_) and participated in all four serological-screening campaigns that followed the completion of the primary cycle (T_1_, T_2_, T_3_, and T_4_) were included in the study. Whenever available, the results of serological screenings performed during 2020 were cumulated with the baseline (T_0_) and used to identify all pre-vaccine infections. Infections occurring during the observational period were detected using rhino-pharyngeal swabs (RPS) that were performed routinely (fortnightly/monthly) or on request in case of contact or symptoms. HCWs who were infected in the 14 days following the last serological test result (T_4_), who could have had altered the antibody levels measured at T_4_ leading to misclassification, were excluded from the analysis. No patients or public were involved in the study design.

### 2.2. The Cohort

Overall, 4824 vaccinated workers fulfilled the inclusion criteria; all the included subjects were vaccinated (primary cycle) with the BNT162b2 vaccine and had all the test results of the T_0_, T_1_, T_2_, T_3_, and T_4_ serological-screening campaigns (no missing data). The third vaccine dose (booster, a mRNA vaccine) was administered between T_3_ and T_4_. For each participant, the starting date of the observational period coincided with the date of the blood sample collection for the serological assay performed at T_4_. All participants were monitored up to 31 October 2022. A total of 2863 workers who did not show records of anti-N seroconversion induced by natural infection were considered pure-vaccinated, and their serological response (anti-S antibody levels) was used as a reference for comparison, while those with evidence of SARS-CoV-2 infection (anti-N seroconversion and pre-vaccine anti-S seroconversion) were considered hybrid-immunized workers. In such cases, the specific SARS-CoV-2 variant responsible for natural infections was defined according to the records contained in the sero-surveillance programme and the information available at the European Centre for Disease Prevention and Control as follows: pre-vaccine infections (N = 986, *wild strain*), post-vaccine infections (N = 75, *non-omicron strains*), and peri-booster infections (N = 900, *omicron strains*) [20].

### 2.3. Serological Assays

During spring 2020, serum samples were tested using the chemiluminescent immunoassay Liaison^®^ SARS-CoV-2 S1/S2 IgG assay (DiaSorin^®^, Saluggia, Italy), whereas during autumn 2020, electrochemiluminescence immunoassay (ECLIA) Elecsys^®^ Anti-SARS-CoV-2, which detects immunoglobulins (IgG/A/M) anti-N (Roche^®^ Diagnostics International Ltd., Rotkreuz, Switzerland), was used. The response to the vaccine (from T_1_ onward) was assessed using ECLIA Elecsys^®^ Anti-SARS-CoV-2 S for anti-S (IgG/A/M) detection (Roche^®^ Diagnostics International Ltd., Rotkreuz, Switzerland). Liaison^®^ SARS-CoV-2 S1/S2 IgG is a CLIA assay for the in vitro quantitative detection of IgG anti-S (anti-S1 and anti-S2) in serum and plasma. Recombinant S1 and S2 antigens bound to magnetic beads and the mouse monoclonal antibody anti-human IgG were used to detect and quantitate IgG in human samples. The results are expressed as U/mL, and specimens are considered negative if <12 U/mL, equivocal between 12 and 15 U/mL, and positive if ≥15 U/mL. Elecsys^®^ Anti-SARS-CoV-2 is an ECLIA immunoassay for the in vitro qualitative detection of antibodies (IgG/A/M) against SARS-CoV-2 in human serum and plasma. The assay uses a recombinant protein representing the nucleocapsid (N) antigen in a double-antigen sandwich-assay format. The results are expressed as the cut-off index, with the cut-off being 1. Elecsys^®^ Anti-SARS-CoV-2 is an immunoassay for the in vitro quantitative determination of antibodies (IgG/A/M) to the SARS-CoV-2 Spike (S) protein-receptor-binding domain (RBD) in human serum and plasma. The assay uses a recombinant protein representing the RBD of the S antigen in a double-antigen sandwich-assay format. The results are expressed as U/mL, the cut-off was 0.8 U/mL, and the upper limit of detection was 250 U/mL. Since the antibody titres elicited in immunized individuals were very high, we tested all serum samples at a dilution of 1:20, in accordance with Roche, so the upper limit of detection increased to 5000 U/mL, and the dynamic range could be extended.

### 2.4. SARS-CoV-2 Detection in Swabs

Rhino-pharyngeal swabs (RPS) were routinely used to perform fortnightly screening on our personnel as well as to test symptomatic workers and close contacts of a confirmed COVID-19 case. In the case of positive antigenic swabs (cassette), a molecular test was performed to confirm the case. The SARS-CoV-2 ELITe MGB Kit^®^ (Elitechgroup, Turin, Italy) was used for the detection of SARS-CoV-2 virus through reverse transcription (RT) followed by real-time PCR from RNA extracted from RPSs. One-step RT-real-time polymerase chain reaction was used to confirm the presence of COVID-19 by amplification of two regions: the RdRp and ORF8 genes. Extraction, detection, and quantification were performed using a commercial automatized platform (Elite InGenius^®^, Elitechgroup, Turin, Italy). Briefly, primary RPS samples were loaded directly and processed on the Elite InGenius^®^ system according to the manufacturer’s instructions, and the results were available after a 2.5 h process of 200 μL for each sample. Result interpretation and analysis were automatically performed by the Elite InGenius^®^ system.

### 2.5. Statistical Analysis

Continuous variables are presented as medians and interquartile ranges (IQRs) and were compared with the use of the Mann–Whitney test; pairs were compared with the use of the Wilcoxon signed-rank test.

A multivariable piecewise linear mixed-effect regression model was applied to estimate the anti-S decay over time before and after the booster. The model was adjusted for pre-vaccine SARS-CoV-2 infection, gender, and age to address potential sources of bias. Covariates were included based on the hypothesis that they could have an influence on the anti-S trajectories. To test differences among the considered groups, an interaction term between time and the grouping variable (defined as no previous infection, pre-vaccine infection, post-vaccine infection, or peri-booster infection) was introduced. A restricted cubic spline with three knots, set at the pre-booster 1st quartile, at the booster date and at the post-booster 3rd quartile, was applied since anti-S levels showed a nonlinear decay over time. Because of the asymmetric outcome distribution, a total of 500 bootstrap iterations were used. All the included subjects had all the test results of the T_0_, T_1_, T_2_, T_3,_ and T_4_ serological-screening campaigns (no missing data).

Individual data were employed to perform a survival analysis. A failure event was defined as SARS-CoV-2 infection after the T_4_ sampling, as detected by RPS swabs. Survival time—defined as time without infection—was estimated since the booster injection date (i.e., booster dose date was considered time 0).

Kaplan–Meier survival curves were used to graphically represent the survival functions among all the categorical predictors. A log-rank test of equality was performed for each predictor to test the difference in survival (protection/resistance against infection) among the different groups.

Proportional hazard (PH) assumptions were tested based on Schoenfeld residuals and additional graphic methods (it was verified whether −ln−ln (survival) curves for each category of predictors were parallel when plotted vs. ln (analysis time)). The tests revealed that the PH assumption was not met for some of the predictors. This result led to the choice of a parametric framework for the subsequent multivariate survival analysis.

The chosen parametric model for the multivariate analysis was the accelerated time failure (AFT) model, where gender, age, job title, and immunization profile were the selected explanatory variables. In contrast to the proportional-hazards model, an AFT model assumes that the effect of a covariate is to accelerate or decelerate the failure time by some constant.

In this model, the survival time is assumed to follow a known distribution. Thus, the first step was to identify the distribution underlying the AFT model that best fitted the data, as it was the most appropriate according to a comparison analysis based on Akaike’s information criteria (AIC) and Bayesian information criteria (BIC). The goodness-of-fit of the generalized gamma model was evaluated using the Cox–Snell residuals. Finally, the model was used to estimate the effect of the covariates on the survival time, which were expressed in time ratios (also called acceleration factors). The time ratio (TR) for a given covariate is the exponent of the corresponding coefficient: a time ratio greater than 1 implies a longer mean/median survival time (a higher resistance to the infection) and vice versa.

All tests were two-sided, and the statistical significance was set at α = 0.05. Analyses were performed through R (version 4.2.2) and Stata/SE (version 17.0).

## 3. Results

A total of 4824 subjects were included in the analysis and classified according to their immunization profile (Table 1). Each of them was followed starting from the date of the serological test performed at T_4_ (between February and March 2022) up to the 31 October 2022. The longitudinal follow-up of anti-S antibody level and the infection recorded during the observational period were parametrized on the booster injection date, which was set as the reference for time.

### 3.1. Antibody Levels

Values obtained after the primary cycle and before the booster were compared with those obtained after the booster; except for the “pre-vaccine infection group”, in which T_4_ antibody levels were not higher than those observed at T_1_, higher antibody levels (positive ranks) were always observed (Wilcoxon *p* < 0.001). The booster dose was effective in increasing the antibody levels; a noticeable difference was found in the median and mean antibody levels reached after the booster administration (at T_4_) in comparison with those observed at T_3_, right before the booster-dose administration (Table 1).

The median antibody level of the peri-booster infection group at T_3_ was significantly lower than that shown by the no previous infection group at the same point (Mann–Whitney, *p* = 0.014); only the former had seroconverted at the following serological screening (anti-N seroconversion).

The decay trend already reported [5] persisted even after the administration of the booster dose (Appendix A). The role of the specific immunization profile was later considered in the analysis of the serological response induced by vaccination. Such analysis showed significant differences at the different stages both as anti-SARS-CoV-2-S antibody levels and the gradient (Appendix A). In particular, workers with peri-booster infection (omicron strain) and pre-vaccine infection (wild strain) showed no decay in the trend of the humoral response estimated up to the 25th week following the booster administration (Figure 1 and Appendix A). Indeed, comparing the response to the booster and to the primary cycle, an even steeper drop in antibody levels was found in the no-infection group after the booster administration (Appendix A).

### 3.2. SARS-CoV-2 Infection Detected by Anti-N Aeroconversion up to T_4_

Considering seroconversions occurred after vaccination, the rates of infection caused by the omicron strains, which occurred between T_3_ and T_4_ (peri-booster infection group, 900/3763, 24%), were noticeably higher (12×) than those caused by non-omicron strains between T_0_ and T_3_ (post-vaccine infection group, 75/3838, 2%) in a roughly comparable amount of time (7–9 months).

### 3.3. SARS-CoV-2 Infection Detected during the Observational Period (RPS)

A total of 1250 cases of infection were revealed by RPS performed (event = 1) during the observational period, which started for all the included workers on the day of the last serological test performed (T_4_, between February and March 2022). All the other subjects were censored (event = 0); the whole cohort was followed up to the 31 October 2022. The infections recorded during the observational period were parametrized on the booster injection date. The longest follow-up time recorded was 14 months after the booster dose (further descriptive statistics are available in Appendix A). On average, a new infection was recorded between 7 and 8 months after the booster injection (Appendix A).

The highest percentages of infection were registered in subjects with no hybrid immunization (34.3%) and in the post-vaccine infection group (29.3%). The log-rank test showed a significant difference in time to infection among the different groups of subjects defined by their specific immunization profiles (Appendix A). In particular, the reference group (no previous infection) showed a lower resistance to the SARS-CoV-2 infection during the observational period (Figure 2, Table 2). Significant differences in resistance to the infectionwere also observed when the groups were defined based on gender, age, or job title, where females, younger individuals, and nurses showed a higher infection incidence rate (Appendix A).

Using the AFT model with generalized gamma regression, we observed a decreased survival time among females and nurses and other HCWs compared to administrative staff, while an increased survival time was shown by elderly subjects, external workers, and those who were infected by the wild type and the omicron strains (Table 2). The goodness-of-fit of the model was evaluated using the Cox–Snell residuals (see the plot in Appendix A).

## 4. Discussion

The booster dose proved to be effective in increasing the anti-S antibody levels among all the considered groups. In accordance with our findings, a study on the durability of the immune response after COVID-19 booster showed that the waning of antibody level after the third dose of vaccine was slower than that following the completion of the primary cycle [21]. This observation is consistent with those from other studies, which found that increasing doses of anti-HBV vaccine increased the specific antibody levels and their longevity [22,23,24]. The steeper drop in antibody levels observed in the no previous infection group after the booster administration in comparison with those observed after the primary cycle, followed the reaching of very high antibody levels, but it also indicated that very high antibody levels hardly persist in pure vaccinated individuals.

The comparison among different immunization profiles showed that hybrid immunization enhances the response to the vaccine, eliciting higher antibody levels. According to previous studies [9], the longitudinal follow-up of the anti-S antibody levels confirmed that different immunization patterns can produce different humoral response kinetics. As the median antibody level of the peri-booster infection group at T_3_ was lower than that shown by the no-infection group at the same point, and in the T_3_–T_4_ interval, only the former were infected, antibody levels could confer a certain degree of resistance to the infection. The close relationship between the immunization profiles and antibody levels measured in the considered groups did not allow us to weight the specific role of such factors in modifying resistance to SARS-CoV-2 infection. Hybrid immunization does not always provide better protection against infection. In fact, a higher resistance (significantly longer survival period) to reinfection was observed only among the pre-vaccine infection and peri-booster infection groups. Despite the high antibody levels measured at T_4_, such resistance was not observed in the post-vaccine infection group, thus suggesting that the immunization profile plays a major role as a virus–host interaction modifier.

An assessment of the risk of infection should consider the specific epidemiologic moment as well as the differences existing between dominant circulating virus strains and the immunization profile of the examined case, which should include the most complete natural infection records. In the case of missing virus genotyping details, the publicly available information could be used to define the immunization profile of the examined case.

Regardless of the antibody levels measured, pure-vaccinated individuals should always be considered at higher risk of infection in comparison to hybrid-immunized ones.

A significantly higher rate of infections (12×) was observed among the peri-booster infection group (omicron strains, 24%) in comparison to the post-vaccine infection group (non-omicron strains, 2%). The very small number of workers included in the latter represents proof of the vaccine’s effectiveness in preventing infection from non-omicron strains in the first 8 months following vaccination. This amount of time is also consistent with the average time measured for recording the infection after the booster injection.

The post-vaccine infection (occurred after the primary cycle) proved to affect the immune response, reducing the post-vaccine antibody levels [9], it was also related to a to a steeper drop of the antibody levels after the booster as well as to a higher rate of infection in the observational period (from T_4_ to the 31 October 2022) in comparison with the pre-vaccine infection and the peri-booster infection groups. The very first weeks after the first dose of vaccine administration can therefore be considered a vulnerable period, where preventive measures should not be lowered. This could reasonably apply to vaccines other than those for COVID-19 and should be considered in the case of future in-mass vaccination. The effectiveness of the BNT162b2 vaccine first dose was investigated in a large population, where a cumulative risk of 0.57% was found for days 1–12 and 0.27% for days 13–24, with a significant decrease in the risk of infection registered from day 18 after the first dose [25]. Smaller figures were found for symptomatic COVID-19 infections. The longevity of BNT162b2 effectiveness was debated in a study where the ability to provide excellent protection in the initial weeks after vaccination was confirmed, next to a progressively increasing risk of infection after at least 90 days from the second vaccine dose [26].

The peri-booster infection group showed the best risk profile during the observational period, but the number of infection remained not-neglectable (50/900, 5.6%). This finding further highlighted the importance of preventive measures, even in cases of very high antibody levels or recent infection. The exact timing of seroconversion observed in this group during the T_3_–T_4_ interval remains unknown. Reasonably, infections both before and after the booster administration did occur. While the former can be considered the result of waning immunity, an infection occurring after the booster administration can be explained by the emerging of new virus strain carrying structural changes that allow the escaping from the host’s immune system. Finally, when tackling pandemics caused by viruses such as SARS-CoV-2, which can rapidly mutate, approving and producing updated vaccines on a large scale in a short amount of time seems to be essential.

## 5. Conclusions

(i)The booster dose is effective in increasing the antibody levels in all considered subgroups, but very high antibody levels hardly persist in pure vaccinated individuals.(ii)The humoral response decays over time.(iii)Hybrid immunization modifies the decay and can provide better protection.(iv)Such a phenomenon could be related to both the immunization schedule and/or the genomic relationship characteristics of the circulating virus, which should be considered when assessing the risk of infection.(v)The antibody level plays a minor role; the exact grade of protection conferred by the antibody level cannot be defined.

## 6. Limitations and Strengths of the Study

This study does not consider participants’ medical conditions, which can contribute to explain the different susceptibility to the infection observed among the considered groups.

The results of this study are based on the analysis of the infections revealed by RPS, which sensibility and specificity can be affected by several factors [4]. A further (T_5_) serological campaign at the end of the observational period would have allowed us to confirm the results obtained, but we could not perform it.

The number of individuals included in the post-vaccine infection group is small in comparison to the other study’s groups, and it can be seen as a possible limitation to the study.

To the best of our knowledge, this is the first study focusing on the specific immunization schedule of a such large cohort of vaccinated HCWs, which was followed during two years with an intense testing activity. The unprecedented number and frequency of serological tests performed on almost 5000 individuals allowed to define the immunization profile of each worker with optimal reliability. The sample size of the cohort, its age heterogeneity, and the duration of follow-up allow to generalize the observed results to similar populations.

## Figures and Tables

**Figure 1 vaccines-11-00746-f001:**
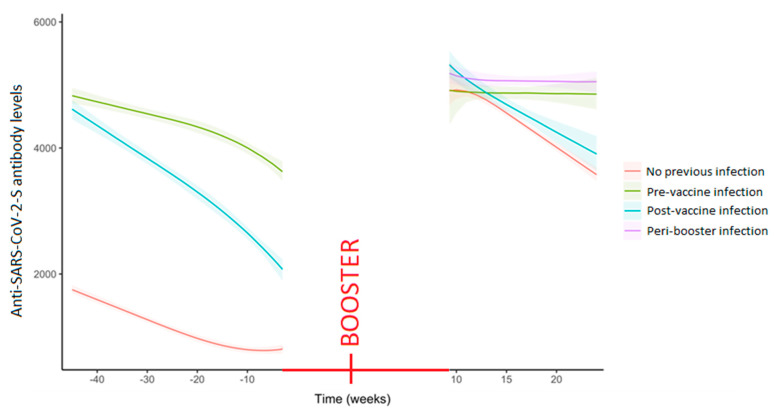
Trends of the anti-SARS-CoV-2-S levels (U/mL) over time in the whole sample stratified by the timing of the occurrence of previous SARS-CoV-2 infection. Curves were obtained from the predictions of the bootstrapped piecewise linear mixed model adjusted by age and gender. The day of the booster injection was set as the reference for time. The gap between the two periods (before and after the booster) was due to the absence of an anti-S serological assay performed during the interval between T_3_ and T_4_, when the booster dose was administered. Due to the infections occurred in such interval the no previous infection group was split in two, originating the peri-booster infection group, which curve can be only seen on the right part of the figure.

**Figure 2 vaccines-11-00746-f002:**
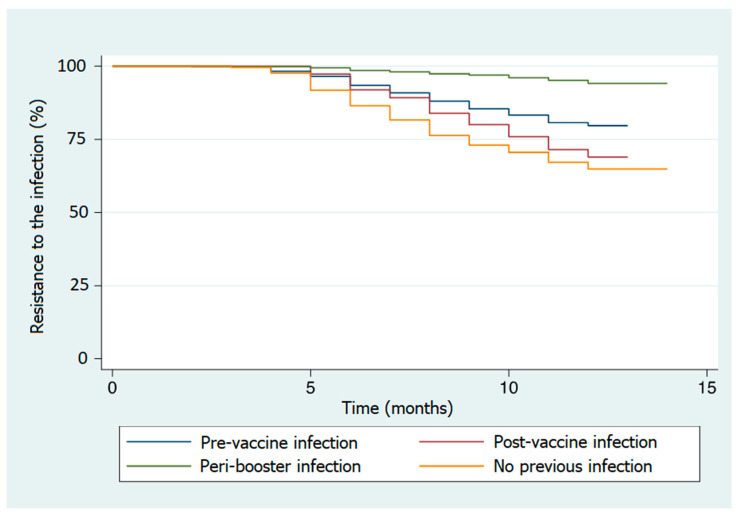
Kaplan–Meier curves measuring time without infection obtained according to the considered immunization profiles. The booster injection date was considered a reference for the time axis.

**Table 1 vaccines-11-00746-t001:** Comparison of the median antibody levels measured at different times among the cohort classified according to the considered immunization profiles.

Immunization Profile (N.)	Natural Infection Period	Circulating Virus Strain	Anti-SARS Ig Levels, Median (IQR)—U/mL		
			After Primary Cycle (T_1_)	Before Booster (T_3_)	After Booster (T_4_)
No previous infection (2863)	/	/	1174 (690–1894)	625 (367–1031)	5000 (4210–5000)
Pre-vaccine infection (986)	20 March 20 December	Wild	5000 (5000–5000)	4071 (2065–5000)	5000 (5000–5000)
Post-vaccine infection (75)	21 May 21 October	Non-omicron	1528 (666–2616)	1744 (678–3442)	5000 (5000–5000)
Peri-booster infection (900)	21 November 22 March	Omicron	1125 (649–1889)	575 (340–972)	5000 (5000–5000)
Whole cohort	20 March 22 March	n.a.	1470 (783–3124)	769 (420–1601)	5000 (5000–5000)

**Table 2 vaccines-11-00746-t002:** Results of the multivariate analysis obtained using an AFT model. Gender, age, job title, and immunization profile were the selected explanatory variables. Time ratio (TR) indicates that a variable can accelerate (TR < 1) or decelerate (TR > 1) the SARS-CoV-2 infection.

	Positive RPS, N. (%)	TR (95% CI)	*p*
Gender			
Male	257/1194 (21.5)	Reference	
Female	993/3630 (27.4)	0.91 (0.85–0.97)	0.005
Age			
20–29	157/562 (27.9)	Reference	
30–39	229/875 (26.2)	1.03 (0.93–1.13)	0.601
40–49	355/1297 (27.4)	1.08 (0.98–1.18)	0.124
50–59	453/1767 (25.6)	1.19 (1.08–1.30)	<0.001
Over 60	56/323 (17.3)	1.36 (1.19–1.56)	<0.001
Job title			
Administrative	135/591 (22.8)	Reference	
Technician	110/406 (27.1)	0.98 (0.87–1.10)	0.691
Other HCW	245/870 (28.2)	0.95 (0.86–1.05)	0.311
Nurse	533/1753 (30.4)	0.88 (0.81–0.97)	0.008
Physician	215/980 (21.9)	1.03 (0.93–1.13)	0.625
External worker	12/224 (5.4)	1.73 (1.45–2.06)	<0.001
Immunization profile			
No previous infection	983/2863 (34.3)	Reference	
Pre-vaccine infection	195/986 (19.8)	1.38 (1.29–1.48)	<0.001
Post-vaccine infection	22/75 (29.3)	1.15 (0.93–1.42)	0.205
Peri-booster infection	50/900 (5.6)	2.26 (2.07–2.47)	<0.001

## Data Availability

Data subject to third-party restrictions: Data are available from the authors, with the permission of the tertiary hospital ASST Spedali Civili di Brescia.

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
