# Peer review of "The Effect of the Immunization Schedule and Antibody Levels (Anti-S) on the Risk of SARS-CoV-2 Infection in a Large Cohort of Healthcare Workers in Northern Italy"

_vaccines, 2023, doi:10.3390/vaccines11040746_

Round 1

Reviewer 1 Report

I appreciate your study, manuscript and conclusions.  Your observations regarding the protection of immunizations and hybrid immunity is appreciated and very valuable.  Additionally, I appreciate your finding that the peri-vaccine periods are times of high susceptibility for vaccinees. 

1.  Recommend the use of generic names versus trade names like Comirnaty

2. When describing the way that your data fit your model you use the term "goodness of fit" a few times.  This is clunky language and I would recommend different language.

3.  Within the text of the article, when you reference your tables and figures, they should say Table 1 or Figure 1 versus table 1.

4.  Your Kaplan-Maier curves you have the Y axis as Survival.  Is it really survival? Isn't more like protection versus infection?

5.  In line 337, you note that the finding is not neglectable. Instead of using the negative, you should say that it is significant versus not neglectable.

6.  You have fonts in your manuscript that switch back and forth, please standardize the fonts.

Author Response

Dear reviewer,
Thank you for your valuable comments. 

  1. An exthensive grammar and lexicon editing has been performed. The draft has also been revised and updated with some missing parts (Abstract and Graphical abstract).
  2. “Goodness of fit” is a technical term currently in use, its substitution does not seem feasible.
  3. The updated draft includes the modifies you suggested.
  4. The survival analysis was the instrument used to study the SARS-CoV-2 infection after the T4 sampling detected by RPS swabs (and which was defined as the failure event). Indeed, the survival time estimated can be seen as protection or resistance versus susceptibility to the infection.
  5. Thank you for your comment, which was received and considered with great interest. Possibly recalling statistics, the word “significant” could be misleading. The negative sentence was therefore preferred.
  6. The draft has now been revised, an extensive grammar and lexicon editing has been performed. The updated draft has been uploaded.

Reviewer 2 Report

The results present clearly in the main text and the conclusion is constant with the other research studies. I only have one comment on the writing of the abstract.

1. The abstract is not clearly presented. Actually, we can not find any information about the green rectangle and green hexagons in the main text. What is their meaning? The author may need to rewrite the abstract, including background, scientific question, and conclusion.

Author Response

Thank you for your valuable comment. I totally agree. The abstract you read should have been instead the Graphical Abstract’s caption, which was probably cut during uploading/editing of the draft. The draft has now been revised; an extensive grammar and lexicon editing has been performed. The updated draft version has been uploaded.

Reviewer 3 Report

The artical raises the important issue of the level of immunity after vaccination in healthcare workers. The results come from a larger-scale observation. However, there are some unclear points that need improvement:

1. I can't find a graphical abstract as well as keywords

2. The abstract is incomprehensible to me.

3.The time scheduke of the study should be written in more detail, with T0, T1, T2, T3 and T4 clearly defined. Is T0 the period when the vaccine was not yet available?

4. From the presented results, it is not possible to conclude whether the authors of the study took into account the type of vaccine (mRNA or vector vaccine?)

5. There are some text fragments that should be edited (e.g. in conclusion lines 352-356: these phrases refer to the sentence point "iii"?), because they are incomprehensible

6. Position 2 in the literature - what is it? Please enter the full name of the data source

Author Response

Thank you very much for your valuable comments.

  1. I apologise for the inconvenience. The abstract you read should have been instead the Graphical Abstract’s caption, which was probably cut during uploading or very first editing of the draft. The draft has now been revised; an extensive grammar and lexicon editing has been performed. The updated draft version has been uploaded.
  2. Please see the above response.
  3. Thank you very much for your comment. This paper updated the results obtained within our serosurveillance programme whith the last results collected. A detailed explanation how the timing was provided with the previous paper (reference n.5), and it is also resumed in the graphical abstract. Please verify that the graphical abstract can provide a sufficient level of information to the reader. Any further comment will be considered with great interest.  
  4. All the subjects included in this analysis received 2 doses of the BNT162b2 vaccine (mRNA vaccine); booster administered was also a mRNA vaccine (BNT162b2 – N. 4081 - or mRNA-1273 – N. 743).
  5. The updated draft includes the modifies you suggested.
  6. The updated draft includes the modifies you suggested.